# Hot Deformation Behavior and Processing Maps of Pure Copper during Isothermal Compression

**DOI:** 10.3390/ma16113939

**Published:** 2023-05-24

**Authors:** Tiantian Chen, Ming Wen, Hao Cui, Junmei Guo, Chuanjun Wang

**Affiliations:** 1Kunming Institute of Precious Metals, Kunming 650106, China; ctt@ipm.com.cn; 2State Key Laboratory of Advanced Technologies for Comprehensive Utilization of Platinum Metals, Sino Platinum Metals Co., Ltd., Kunming 650106, China; wen@ipm.com.cn (M.W.); cuihao@ipm.com.cn (H.C.); gjm@ipm.com.cn (J.G.)

**Keywords:** isothermal compression, pure copper, flow stress, constitutive model, processing map

## Abstract

In this study, pure copper’s hot deformation behavior was studied through isothermal compression tests at deformation temperatures of 350~750 °C with strain rates of 0.01~5 s^−1^ on a Gleeble-3500 isothermal simulator. Metallographic observation and microhardness measurement were carried out of the hot compressed specimens. By analyzing the true stress–strain curves of pure copper under various deformation conditions during the hot deformation process, the constitutive equation was established based on the strain-compensated Arrhenius model. On the basis of the dynamic material model proposed by Prasad, the hot-processing maps were acquired under different strains. Meanwhile, the effect of deformation temperature and strain rate on the microstructure characteristics was studied by observing the hot-compressed microstructure. The results demonstrate that the flow stress of pure copper has positive strain rate sensitivity and negative temperature correlation. The average hardness value of pure copper has no obvious change trend with the strain rate. The flow stress can be predicted with excellent accuracy via the Arrhenius model based on strain compensation. The suitable deforming process parameters for pure copper were determined to be at a deformation temperature range of 700~750 °C and strain rate range of 0.1~1 s^−1^.

## 1. Introduction

Pure copper is widely used in integrated circuits due to its low resistivity, high electromigration resistance, high thermal conductivity, and corrosion resistance. In the manufacturing process of modern integrated circuits, sputtering is the main method of film formation. Compared with vacuum evaporation coating, it has the advantages of high energy of sputtered particles, high compactness, and high controllability of film quality [1]. Therefore, with the development of integrated circuits, the demand for thin film material is booming, which also leads to strict requirements for various properties of sputtering targets. At present, ultra-high purity copper target is the key raw material for sputtering of copper interconnects seed layer and packaging application [2]. High-quality sputtered films could be obtained by well-controlled microstructure and crystallographic texture of sputtering targets, which are determined by the working process. Hot deformation, including as hot forging and hot rolling, are essential for achieving regulated grain size and texture. These techniques may also be used to seal pore or void defects that were originally present in the as-cast ingot. To accurately describe the impact of process variables, such as temperature, deformation degree, and strain rate, on the flow stress as well as to provide a solid foundation for the processing technology of pure copper target, it is crucial to study the hot deformation behavior of pure copper. There are some reports on pure copper’s hot deformation behavior. At temperatures ranging from 523 to 773 K and strain rates of 10^−4^~10^−1^ s^−1^, Gao et al. [3] investigated the hot deformation behavior and associated structure changes of polycrystalline copper with various purities. The activation energy was measured to be between 210 and 245 kJ/mol. At the deformation temperatures between 843 and 993 K and strain rates between 10^−3^ and 10 s^−1^, Yang et al. [4] studied the hot compression behavior of pure copper. The activation energy was 215 kJ/mol, showing a lattice-diffusion-controlled plastic flow behavior. At temperatures between 400 and 900 °C and strain rates between 0.001 and 1 s^−1^, Huang et al. [5] studied the strain-hardening and -softening behaviors of pure copper. The flow instability occurred in the zones of 400–450 °C, 0.001–0.05 s^−1^, and 450–750 °C, 0.05–1 s^−1^. However, the evolution of microstructure and hardness during hot compression deformation is rarely studied.

In this study, pure copper was subjected to isothermal compression tests at various deformation temperatures and strain rates. Both the constitutive equation and the hot-processing maps were established. In the meantime, the impact of deformation temperature and strain rate on the evolution of the microstructure, hardness, and plastic deformation was examined. The parameters of the deforming process were then optimized.

## 2. Materials and Methods

The raw material for this study was 4N purity pure copper. Table 1 displays the impurity content of the raw material as determined by ICP-MS. Figure 1 illustrates the optical microstructure of pristine sample of pure copper.

Hot compression experiments were performed on a Gleeble-3500 thermal-mechanical simulator using machined cylindrical specimens with a diameter of 10 mm and a height of 15 mm that were extracted from a forged stock. Figure 2 schematically shows the experimental process for hot compression testing and specific parameters. To guarantee uniform temperature distributions inside, the specimens were heated to the preset deformation temperature at a rate of 5 °C/s and held there for 5 min before deformation. After isothermal compression, the specimens were promptly quenched by cooled water to preserve the high-temperature microstructure for examination.

After deformation, the specimens were cut along the center plane in the compression direction. Optical observation specimens were prepared by etching the observation area with etching solution (5 g ferric chloride, 10 mL hydrochloric acid, and 82 mL absolute ethanol) after standard metallographic operations by a Struers Tegramin-25 device. The longitudinal section’s middle was chosen as the observation location to lessen the impact of uneven deformation [6]. Optical microstructure was observed by a laser confocal microscope (ZEISS Smartproof 5). According to ASTM E112-12, the Abrams three-circle approach was utilized to determine the specimens grain size [7]. Vickers hardness measurements were performed with a 0.1 kg force applied for 10 s using a microhardness tester Shanghai Baoleng HXD-1000TMC/LCD). 

The interface friction between specimen and dies would have an impact on the symmetrical deformation of the specimen [8,9,10]. Despite the experiment’s use of lubricants to lessen interfacial friction, the cylindrical specimens nevertheless had a barrel-shaped appearance or bulged. The contact area between the specimen and dies grows as the temperature and degree of deformation rise, and the interface friction becomes more and more visible, resulting in increasingly uneven deformation of the specimen. Therefore, in order to measure the effectiveness of the unidirectional hot compression test, when the bulge occurs at the waist of the specimen after the hot compression test, the bugle coefficient, *B*, can be provided to evaluate the correctness of the flow stress curve produced during experimentation [11]. The change in specimen size before and after deformation, which could be assessed using the following expression, is used to determine *B*:(1)$$\begin{array}{c} B=L_{0}d_{0}^{2}/L_{f}d_{f}^{2} \end{array}$$
where *L*_0_ and *L_f_* denote the heights of the un-deformed (15 mm) and deformed specimens, respectively. *d*_0_ and *d_f_* are the diameters of the un-deformed (10 mm) and deformed specimens, respectively.

When *B* is more than 0.9, the results of unidirectional hot compression test are effective [11]. As shown in Table 2, the bugle coefficients under different deformation conditions are more than 0.9, indicating the validity of the stress–strain curves produced from all the specimens.

## 3. Results

### 3.1. True Stress–Strain Curves

Figure 3 displays the true stress–strain curves of pure copper obtained from hot compression testing. The flow stress of pure copper exhibits a negative temperature correlation and a positive strain rate sensitivity, which is evident from the observation that the flow stress increases with decreasing deformation temperature as well as rising strain rate. The material during hot deformation undergoes a combination action of work hardening and dynamic softening [12,13]. Figure 3 shows that the flow stress rises with the increasing strain, while the growth trend slows down gradually. The flow stress steadily reduces after reaching its peak and eventually tends to stabilize. The curves of flow stress that keep increasing with increasing strain belong to a dynamic recovery type one. Additionally, as deformation increases, dynamic recrystallization type one occurs, where the flow stress first rises and then falls. During the initial stage, that is, prior to the peak point, the dynamic recovery effects cannot completely counteract the work hardening effects. Because its effects are dominant, the flow stress increases rapidly. The dislocation density also rises as the degree of deformation increases. A significant number of dislocations are removed when dynamic recrystallization takes place under a specific critical deformation condition. Due to the dominant recrystallization softening, the flow stress starts to decrease, and the peak stress point appears. Subsequently, the effects of hardening and softening tend to balance, and the curves become more and more smooth. Moreover, the flow stress tends to enter a stable phase at low rates and high temperatures, which is characterized by dynamic flow softening. 

#### 3.1.1. Microstructure Evolution

Optical microstructure characterization of compressed samples with various deformation parameters was carried out to evaluate the evolution of pure copper’s microstructure under hot deformation (Figure 4 and Figure 5). The original structure is replaced by fine recrystallized grains with an average grain size of 36.6 m when the strain rate is 5 s^−1^ in Figure 4a. The amount of time needed for the specimen to deform to the same degree increases as the strain rate decreases. Dynamic recovery and dynamic recrystallization are entirely accomplished. Thus, the grains have enough time to grow, and the average grain size increases, as shown in Figure 4b–d. When the strain rate decreases to 1 s^−1^, the average grain size reaches 64.8 μm. From Figure 4d, pure copper is fully recrystallized at an elevated temperature of 750 °C, and its grain size reaches 110.1 μm. 

Figure 5a–e display the optical micrographs taken at various temperatures with a strain rate of 0.01 s^−1^. Figure 5a displays that at 350 °C, the grains of pure copper present the deformation state after compression. The grain boundaries are blurred, and the grains show a clearly stripped shape. Under this condition, the isothermal hot compression of pure copper shows dynamic recovery characteristics. When the temperature rises to 450 °C, that is, it is above the theoretical recrystallization temperature, obvious dynamic recrystallization occurs in pure copper. In Figure 5b, the grain boundaries present a sawtooth shape, with fine dynamic recrystallized grains. According to Figure 5c,d, with the increase of deformation temperature, full recrystallization is achieved, and the recrystallized grains grow continuously, even with uneven distribution and abnormal growth. This is because as the temperature increases, the energy provided for dynamic recrystallization is more abundant. It provides a better environment for the growth of recrystallized grains and enhances the migration of grain boundaries; thus, the grain size gradually increases.

#### 3.1.2. Microhardness Analysis

It is commonly acknowledged that the mechanical properties of the material are greatly influenced by its microstructure [14]. The evolution of the microstructure can also be reflected in the hardness change that occurs under various deformation conditions. Figure 6 displays the relation between deformation conditions and Vickers hardness at the center of the longitudinal section of the specimen. Vickers hardness typically decreases as the deformation temperature rises, and the hardness value is more uniform at high temperatures. The hardness trend is not apparent as the strain rate increases. The increase in temperature is conducive to the migration of grain boundaries as well as the movement of dislocations and promotes the softening effects; thus, the hardness reduces. In addition, elevating the temperature increases the grain size. According to the Hall–Petch relationship, larger grain size will lead to the reduction of hardness. As for the non-uniformity of hardness value, the unstable hardness value is due to the incomplete dynamic recrystallization at low temperature, resulting in a big variation in grain size and hardness values in different regions [15]. High-density dislocations are easy to produce as the strain rate rises, but there is not enough time for dynamic recovery or recrystallization softening; therefore, the hardness is high. On the other hand, when the deformation rate is large, the coarse and fine grains are staggered relative to one another. The hardness difference between coarse grain and fine grain is obvious. When the hardness points are randomly selected, the average hardness value decreases. Therefore, there is no discernible pattern in how strain rate affects hardness when these two factors interact.

### 3.2. Establishment and Modification of Constitutive Models of Pure Copper

#### 3.2.1. Subsubsection

The relationship between strain rate, deformation temperature, and flow stress at high deformation temperature is commonly described using the Arrhenius model [16,17]. It can characterize the features of the stress–strain curve that increases initially before dropping. This model is usually denoted by the following three equations:(2)$$\begin{array}{c} \begin{array}{c} \dot{\varepsilon }=A{\left[\sinh\left(\alpha \sigma \right)\right]}^{n}\exp\left(-\frac{Q}{RT}\right)\left(for\;all\;\alpha \sigma \right) \end{array} \end{array}$$
(3)$$\begin{array}{c} \begin{array}{c} \dot{\varepsilon }=A_{1}\sigma^{n_{1}}exp\left(-\frac{Q}{RT}\right)\left(\alpha \sigma <0.8\right) \end{array} \end{array}$$
(4)$$\begin{array}{c} \begin{array}{c} \dot{\varepsilon }=A_{2}exp\left(\beta \sigma \right)exp\left(-\frac{Q}{RT}\right)\left(\alpha \sigma >1.2\right) \end{array} \end{array}$$
where $\dot{\varepsilon }$ is the strain rate (s^−1^). σ is the flow stress (MPa). R is the universal gas constant (8.3145 $J\cdot {mol}^{-1}\cdot K^{-1}$). T is the absolute temperature (K). Q is the activation energy of hot deformation ($J\cdot {mol}^{-1}$). A, $A_{1}$, $A_{2}$, α, n, $n_{1}$, and β are the material constants, and $\alpha =\beta /n_{1}$ [18,19]. 

Taking the logarithms of the two sides of Equations (2)–(4), respectively, gives:(5)$$\begin{array}{c} \begin{array}{c} \ln\dot{\varepsilon }=\ln A+n\ln\left[\sinh\left(\alpha \sigma \right)\right]-\frac{Q}{RT} \end{array} \end{array}$$
(6)$$\begin{array}{c} \begin{array}{c} \ln\dot{\varepsilon }=\ln A_{1}+n_{1}\ln\sigma -\frac{Q}{RT} \end{array} \end{array}$$
(7)$$\begin{array}{c} \begin{array}{c} \ln\dot{\varepsilon }=\ln A_{2}+\beta \sigma -\frac{Q}{RT} \end{array} \end{array}$$

In Figure 7a,b, by linear fitting the $\ln\dot{\varepsilon }-\ln\sigma$ and $\ln\dot{\varepsilon }-\sigma$, and computing the slope value of each fitting curve, $n_{1}$ and β are obtained as 7.3964 and 0.1282, respectively. Then, the value of α can be calculated as 0.01733 MPa^−1^.

Equation (4) can be converted into:(8)$$\begin{array}{c} \ln\left[\sinh\left(\alpha \sigma \right)\right]\begin{array}{c} =\frac{1}{n}\ln\dot{\varepsilon }-\frac{1}{n}\ln A+\frac{Q}{nRT} \end{array} \end{array}$$

Through the analysis of $\ln\left[\sinh\left(\alpha \sigma \right)\right]-\ln\dot{\varepsilon }$, as shown in Figure 7c, the value of *n* can be derived as 4.8906 from the slopes. 

Differentiating Equation (5) gives:(9)$$\begin{array}{c} \begin{array}{c} Q=R{\left[\frac{\partial \ln\dot{\varepsilon }}{\partial \ln\left[\sinh\left(\alpha \sigma \right)\right]}\right]}_{T}{\left[\frac{\partial \ln\left[\sinh\left(\alpha \sigma \right)\right]}{\partial \left(1/T\right)}\right]}_{\dot{\varepsilon }} \end{array} \end{array}$$

The relationships among $\ln\dot{\varepsilon }-\ln\left[\sinh\left(\alpha \sigma \right)\right]$ and $\ln\left[\sinh\left(\alpha \sigma \right)\right]-1/T$ are gained as in Figure 7c,d. According to Equation (9), the thermal activation energy, Q, of pure copper is 202.10 KJ/mol.

Zener and Hollomon [20] proposed Z-parameter to elaborate the effect of deformation temperature and strain rate on flow stress. Z-parameter is given as:(10)$$\begin{array}{c} \begin{array}{c} Z=\dot{\varepsilon }exp\left(\frac{Q}{RT}\right) \end{array} \end{array}$$

According to Equations (2) and (10), Z-parameter can also be represented as follows:(11)$$\begin{array}{c} \begin{array}{c} Z=A{\left[\sinh\left(\alpha \sigma \right)\right]}^{n} \end{array} \end{array}$$

Taking the logarithm of both sides of Equation (11), gives: (12)$$\begin{array}{c} \begin{array}{c} \ln Z=\ln A+n\ln\left[\sinh\left(\alpha \sigma \right)\right] \end{array}. \end{array}$$

By substituting the *Q* value into Equation (12) and drawing the $\ln Z-\ln\left[\sinh\left(\alpha \sigma \right)\right]$ plot, the relation between ln⁡Z and $\ln\left[\sinh\left(\alpha \sigma \right)\right]$ can be obtained, as shown in Figure 8. Through linearly fitting these data, the value of ln⁡A and *n* can be derived as 21.8026 and 4.8248.

As a result, the constitutive equation of pure copper during hot deformation can be designated as:(13)$$\begin{array}{c} \dot{\varepsilon }=e^{21.8026}{\left[\sinh\left(0.01733\sigma \right)\right]}^{4.8248}exp\left(-\frac{202.10\times 10^{3}}{RT}\right) \end{array}$$

The above formula can be converted to:(14)$$\begin{array}{c} \sigma =\frac{1}{0.01733}\ln\left\{{\left(\frac{\dot{\varepsilon }exp\left(202100/RT\right)}{e^{21.8026}}\right)}^{\frac{1}{4.8248}}+{\left[{\left(\frac{\dot{\varepsilon }exp\left(202100/RT\right)}{e^{21.8026}}\right)}^{\frac{2}{4.8248}}+1\right]}^{1/2}\right\} \end{array}$$

#### 3.2.2. Strain-Compensated Arrhenius Constitutive Model

From Figure 3, the flow stress values under different strains are quite different. Therefore, the strain has a considerable impact on the material constants, and the relationship between the material constants and strain can be explained using the polynomial fitting method. The relationship between β, α, n, Q, and ln⁡A can be fitted by strain:(15)$$\begin{array}{c} \begin{array}{c} \left\{\begin{array}{c} \beta =B_{0}+B_{1}\varepsilon +B_{2}\varepsilon^{2}+B_{3}\varepsilon^{3}+B_{4}\varepsilon^{4}+B_{5}\varepsilon^{5}+B_{6}\varepsilon^{6}+B_{7}\varepsilon^{7}+B_{8}\varepsilon^{8}+B_{9}\varepsilon^{9}\\ \alpha =C_{0}+C_{1}\varepsilon +C_{2}\varepsilon^{2}+C_{3}\varepsilon^{3}+C_{4}\varepsilon^{4}+C_{5}\varepsilon^{5}+C_{6}\varepsilon^{6}+C_{7}\varepsilon^{7}+C_{8}\varepsilon^{8}+C_{8}\varepsilon^{8}\\ n=D_{0}+D_{1}\varepsilon +D_{2}\varepsilon^{2}+D_{3}\varepsilon^{3}+D_{4}\varepsilon^{4}+D_{5}\varepsilon^{5}+D_{6}\varepsilon^{6}+D_{7}\varepsilon^{7}+D_{8}\varepsilon^{8}+D_{9}\varepsilon^{9}\\ Q=E_{0}+E_{1}\varepsilon +E_{2}\varepsilon^{2}+E_{3}\varepsilon^{3}+E_{4}\varepsilon^{4}+E_{5}\varepsilon^{5}+E_{6}\varepsilon^{6}+E_{7}\varepsilon^{7}+E_{8}\varepsilon^{8}+E_{9}\varepsilon^{9}\\ \ln A=F_{0}+F_{1}\varepsilon +F_{2}\varepsilon^{2}+F_{3}\varepsilon^{3}+F_{4}\varepsilon^{4}+F_{5}\varepsilon^{5}+F_{6}\varepsilon^{6}+F_{7}\varepsilon^{7}+F_{8}\varepsilon^{8}+F_{9}\varepsilon^{9} \end{array}\right. \end{array} \end{array}$$

Therefore, the strain compensated Arrhenius constitutive equation can be represented as:(16)$$\begin{array}{c} \begin{array}{c} \sigma =\frac{1}{\alpha \left(\varepsilon \right)}\ln\left\{{\left(\frac{\dot{\varepsilon }exp\left[\frac{Q\left(\varepsilon \right)}{RT}\right]}{A\left(\varepsilon \right)}\right)}^{1/n\left(\varepsilon \right)}+{\left[{\left(\frac{\dot{\varepsilon }exp\left[\frac{Q\left(\varepsilon \right)}{RT}\right]}{A\left(\varepsilon \right)}\right)}^{2/n\left(\varepsilon \right)}+1\right]}^{1/2}\right\} \end{array} \end{array}$$

The material constants under various strains were determined within the range of 0.1~0.7, and the interval is 0.1 using the methods given above. Figure 9 illustrates the relationship between material constants and strain obtained by a ninth order polynomial fitting. The polynomial coefficients of material constants of pure copper are shown in Table 3.

#### 3.2.3. Prediction Accuracy Evaluation

The comparison between the experimental and predicted data for the strain-compensated Arrhenius constitutive model is shown in Figure 10. In addition, the predicted result and the experimental one are in good agreement The correlation coefficient (R) and average absolute relative error (*AARE*) are added to further confirm the prediction precision of the constructed constitutive model of pure copper. The following are the corresponding expressions of *R* and *AARE*:(17)$$\begin{array}{c} R=\frac{\sum_{i=1}^{n}\left(E_{i}-\bar{E}\right)\left(P_{i}-\bar{P}\right)}{\sqrt{\sum_{i=1}^{n}{\left(E_{i}-\bar{E}\right)}^{2}}\sqrt{\sum_{i=1}^{n}{\left(P_{i}-\bar{P}\right)}^{2}}} \end{array}$$
(18)$$\begin{array}{c} AARE=\frac{1}{n}\sum_{i=1}^{n}\left|\frac{E_{i}-P_{i}}{E_{i}}\right|\times 100\% \end{array}$$
where *n* is the number of the data points, *E_i_* is the experimental flow stress, *P_i_* is the predicted flow stress, $\bar{E}$ and $\bar{P}$ are the mean values of *E_i_* and *P_i_*. Figure 11 illustrates the relevance between the predicted and experimental values under various deformation conditions. The *R* value and *AARE* value of the strain-compensated Arrhenius model is 0.9763 and 10.5%, respectively. In conclusion, the established strain-compensated Arrhenius constitutive equation has high accuracy in predicting flow stress.

### 3.3. Hot-Processing Maps of Pure Copper

#### 3.3.1. Construction and Analysis of Hot-Processing Maps

The dynamic material model (DMM), which combines physical system modeling and irreversible thermodynamic theory, is based on the basic concepts of continuum mechanics with massive plastic deformation [21]. The processing map of DMM can not only describe microstructure evolution but also distinguish the instability region during hot forming. According to the dissipative structure theory, the total energy, *P*, dissipated into the system is able to be separated into two components and expressed as:(19)$$\begin{array}{c} \begin{array}{c} P=\sigma \cdot \dot{\varepsilon }=G+J=\int_{0}^{\dot{\varepsilon }}\sigma d\dot{\varepsilon }+\int_{0}^{\sigma }\dot{\varepsilon }d\sigma \end{array} \end{array}$$
where *G* is the energy required for plastic deformation. *J* is the energy dissipated during structural deformation. The strain rate sensitivity parameter, *m*, of the material under a specific stress determines the distribution between *G* and *J*. Thus, *m* can be described as:(20)$$\begin{array}{c} \begin{array}{c} m=\frac{\partial J}{\partial G}=\frac{\dot{\varepsilon }\partial \sigma }{\sigma \partial \dot{\varepsilon }}=\frac{\partial \ln\sigma }{\partial \ln\dot{\varepsilon }} \end{array}. \end{array}$$

When *m* = 1, it is the state of linear dissipation, and the value *J* reaches the maximum: (21)$$\begin{array}{c} \begin{array}{c} J_{max}=\frac{P}{2}=\frac{\sigma \cdot \dot{\varepsilon }}{2} \end{array}. \end{array}$$

Only the strain rate influences the flow stress when temperature and strain are constant, as shown by the following expression:(22)$$\begin{array}{c} \begin{array}{c} \sigma =A\cdot {\dot{\varepsilon }}^{m} \end{array}. \end{array}$$

Moreover, *J* can be described as:(23)$$\begin{array}{c} \begin{array}{c} J=\int_{0}^{\sigma }\dot{\varepsilon }d\sigma =\int_{0}^{\sigma }{\left(\frac{\sigma }{A}\right)}^{\frac{1}{m}}d\sigma =\frac{1}{A^{1/m}}\frac{\sigma^{\left(1/m+1\right)}}{\left(1/m+1\right)}=\frac{\dot{\varepsilon }\cdot \sigma \cdot m}{m+1} \end{array}. \end{array}$$

The efficiency of energy dissipation (η), which describes the percentage of energy dissipated during structural transformation, is shown as follows: (24)$$\begin{array}{c} \begin{array}{c} \eta =\frac{J}{J_{max}}=\frac{2m}{m+1} \end{array}. \end{array}$$

The maximal entropy production principle put forward by Zeigler served as the foundation for the instability criterion given by Prasad et al. [22,23]. In light of this, the Prasad criterion is given as follows: (25)$$\begin{array}{c} \begin{array}{c} \xi \left(\dot{\varepsilon }\right)=\frac{\partial \ln\left(\frac{m}{m+1}\right)}{\partial \ln\dot{\varepsilon }}+m<0 \end{array}. \end{array}$$

According to the flow stress under various deformation circumstances, the efficiencies of energy dissipation (η) and plastic flow instability parameters (ξ) at the strain of 0.2, 0.4, and 0.6 were calculated, respectively. In Figure 12 the shadings represent the areas of flow instability with negative values of ξ, and the contour lines represent the effectiveness of energy dissipation. Moreover, the larger value of η indicates that the microstructure has uniform and equiaxed grains, which improves a better hot workability [24,25]. It is evident that the energy dissipation efficiency increases as deformation temperature rises and strain rate falls. The instability zones tend to gradually grow as the strain increases. Table 4 summarizes the range of process parameters corresponding to the unstable regions under different strain conditions, which should be avoided when selecting hot-working parameters.

Figure 12a demonstrates that the instability zone is generally dispersed at the strain of 0.2 and that as temperature rises, energy dissipation efficiency increases. It reaches the peak value in the region within temperature of 625~680 °C and strain rate of 0.01~0.02 s^−1^, and the peak efficiency is 45%, indicating that this region is the optimal processing region of materials when the strain is 0.2. From Figure 12c, when the strain is 0.6, the material has two processing safety regions, and the safety area and instability area are X-shaped, indicating that temperature and strain rate affect the processing characteristics. The energy dissipation efficiency in the region of low temperature and high-strain rate is low, and its value is 11%, indicating that the energy dissipation efficiency achieves its maximum in the zone of high temperature and low-strain rate, with strain rate ranging from 0.01 to 0.2 s^−1^. Therefore, when the strain is 0.6, the suitable processing window for pure copper is at the temperature range of 540~750 °C and strain rate range of 0.01~0.206 s^−1^.

In Figure 12, with increasing strain, the safety area can be transformed into the instability area and vice versa. Thus, the processing map established under a single strain cannot comprehensively and accurately determine the instability area and safety area in the whole hot compression process. It is essential to take into account the processing maps under various strains to guarantee the continuity of the hot-working process and the correctness of the instability area and safety area. To get the processing map of pure copper throughout the entire hot compression process, the processing maps with strains of 0.2, 0.4, and 0.6 were superimposed. There are just three safety regions, as seen in Figure 13, and the majority of the overlay processing maps are covered by instability areas: (1) safety region I: temperature range 420~510 °C; strain rate range 0.85~5 s^−1^; energy dissipation efficiency 11%; (2) safety region II: temperature range 650~750 °C; strain rate range 0.01~0.018 s^−1^; energy dissipation efficiency 24%; (3) safety region III: temperature range 700~750 °C; strain rate range 0.01~1 s^−1^; energy dissipation efficiency 31%. To optimize the hot-deforming parameters of material, not only the safety area with high-power dissipation efficiency should be selected, but also the realizability of deformation conditions in the actual production process should be considered. Based on the above analysis, the safety area I is abandoned because of the low-energy dissipation efficiency, which increases the probability of flow instability. The safety area II is abandoned because the range of strain rate in this area is too narrow and the rate is very low, which is difficult to achieve in actual production.

#### 3.3.2. Optimal Process Interval Analysis

The optical micrographs taken at various deformation conditions in domains A, B, C, and D are represented in Figure 14a–d, respectively. Figure 14a shows the microstructure of pure copper in domain A at 350 °C and strain rate of 5 s^−1^, which is in the plastic instability region. In this area, it is seen that there is a banded structure inside or between the grains; this means that the microstructure exhibits the traits of local flow instability at high-strain rates and low temperatures. Domain B is undergoing deformation at a temperature of 450 °C and a strain rate of 0.1 s^−1^. From Figure 14b, it appears that the instability degree is weakened, and the microstructure distribution is inhomogeneous. Only a small amount of dynamic recrystallization takes place with very tiny recrystallization grains. Figure 14c shows the microstructure in domain C when the deformation condition is 750 °C and 0.01 s^−1^. It is clear that dynamic recrystallization entirely takes place and that the recrystallized grains become coarser. This is due to the fact that dynamic recrystallization is encouraged by high temperature, and the recrystallized grains have time to grow at a low-strain rate, which leads to microstructure coarsening. Figure 14d shows the microstructure in domain D at 750 °C and 1 s^−1^ deformation parameter. With an average grain size of 53 μm, the central region is mainly composed of uniformly sized, equiaxed crystals. According to the processing map, the energy dissipation efficiency in this area is high and close to the peak value, indicating that the processing parameters can ensure that a high proportion of energy would be used for the microstructure evolution. Thus, this area’s processing parameters are suitable for actual production. Therefore, the preferred hot deformation condition of pure copper should be within the temperature range of 700~750 °C and strain rate range of 0.01~1 s^−1^.

## 4. Conclusions

Based on hot compression testing, constitutive equations and processing maps of pure copper were constructed. The effects of deformation conditions on the flow behavior and microstructure evolution of pure copper were analyzed. The conclusions that can be made are as follows:(1)Pure copper features a flow stress that is negatively correlated with temperature and positively sensitive to strain rate. With an increase in deformation temperature and a decrease in strain rate, pure copper’s average grain size grows. The average hardness value of pure copper declines with rising temperature but has no obvious change trend with the strain rate.(2)The constitutive equation for pure copper was established via the strain-compensated Arrhenius model. The value of the correlation coefficient and the average absolute relative error values were 0.9763 and 10.5%, respectively, showing great accuracy in predicting flow stress, which can provide a reference for practical production and numerical simulation.(3)Based on the DMM model, the hot-processing maps were constructed. The suitable processing window of pure copper was determined at the temperature range of 700~750 °C and strain rate range of 0.1~1 s^−1^.

## Figures and Tables

**Figure 1 materials-16-03939-f001:**
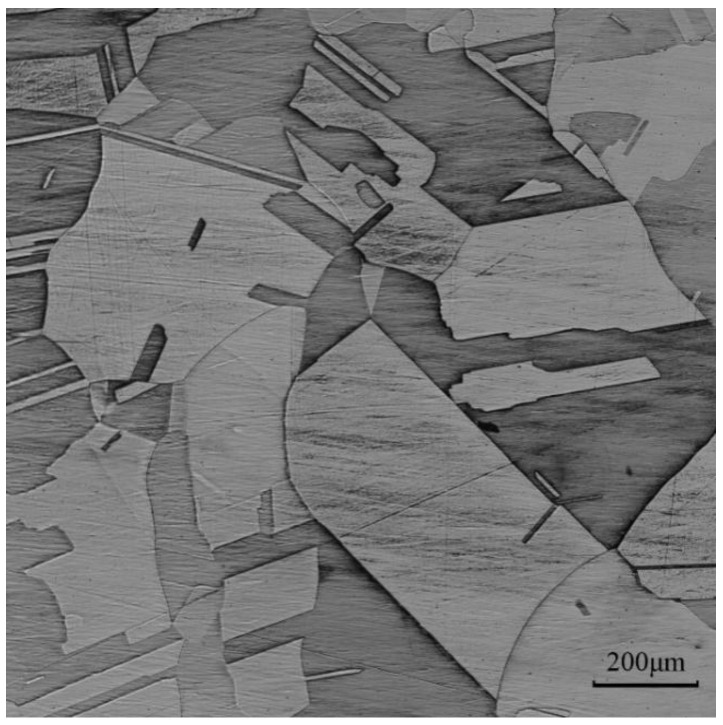
Initial microstructure of pristine sample of pure copper.

**Figure 2 materials-16-03939-f002:**
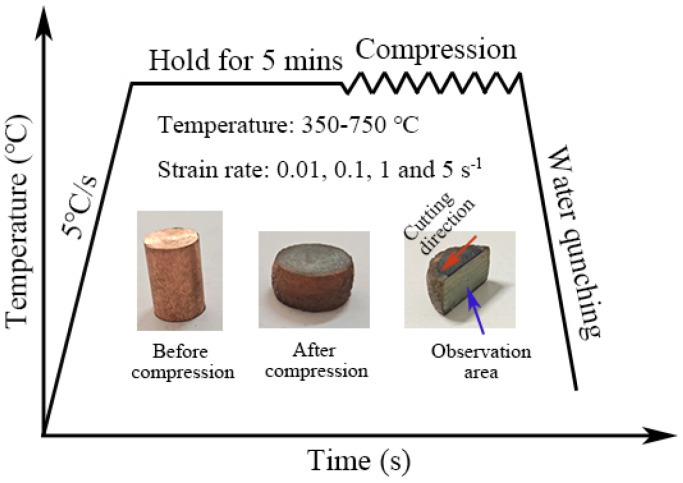
Schematic diagram of hot compression tests and specimen morphology.

**Figure 3 materials-16-03939-f003:**
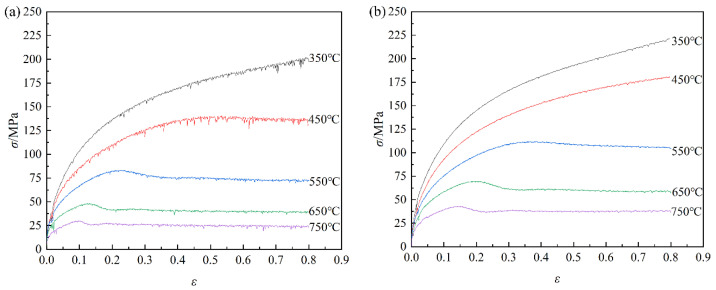
True stress–strain curves of pure copper under various deformation conditions: (**a**) *έ* = 0.01 s^−1^; (**b**) *έ* = 0.1 s^−1^; (**c**) *έ* = 1 s^−1^; (**d**) *έ* = 5 s^−1^.

**Figure 4 materials-16-03939-f004:**
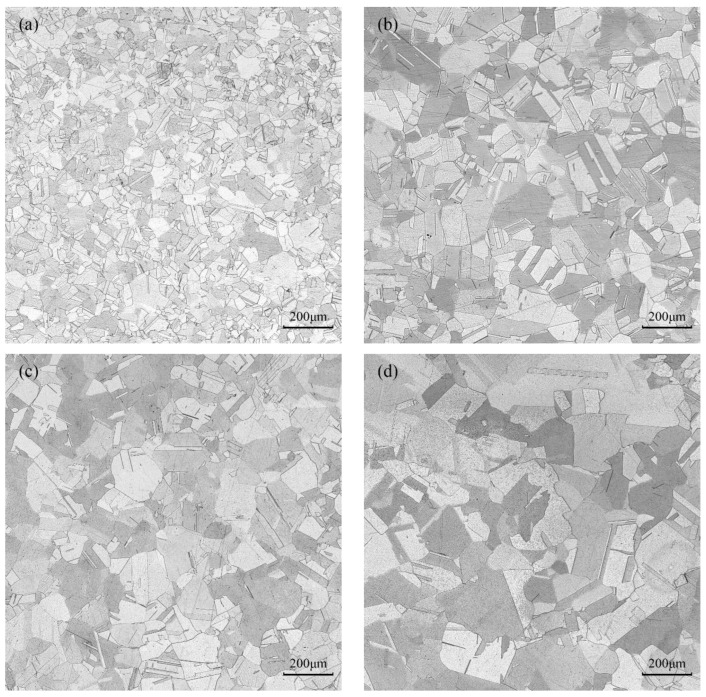
Optical micrographs of pure copper deformed at the same temperature of 750 °C with various strain rates: (**a**) *έ* = 5 s^−1^; (**b**) *έ* = 1 s^−1^; (**c**) *έ* = 0.1 s^−1^; (**d**) *έ* = 0.01 s^−1^.

**Figure 5 materials-16-03939-f005:**
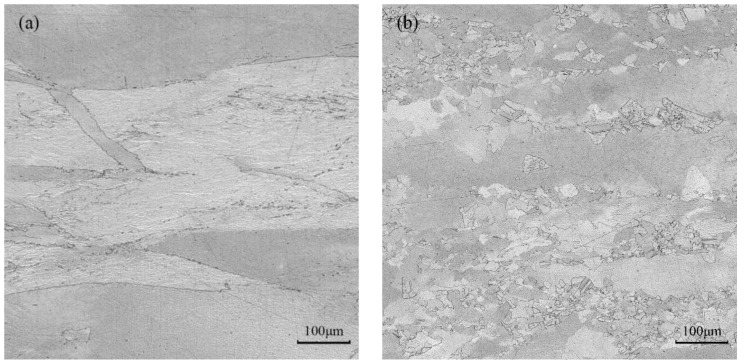
Optical micrographs of pure copper at the same strain rate of 0.01 s^−1^ with various temperatures: (**a**) *T* = 350 °C; (**b**) *T* = 450 °C; (**c**) *T* = 550 °C; (**d**) *T* = 650 °C; (**e**) *T* = 750 °C.

**Figure 6 materials-16-03939-f006:**
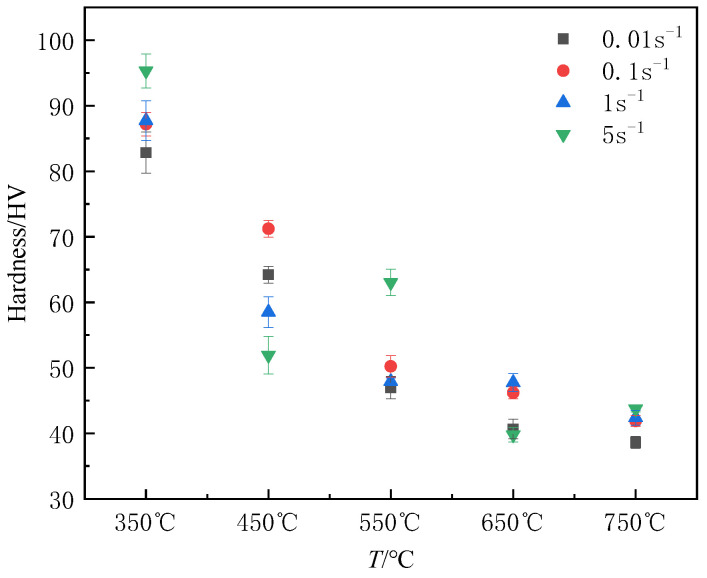
Vickers hardness of pure copper at different hot compression conditions.

**Figure 7 materials-16-03939-f007:**
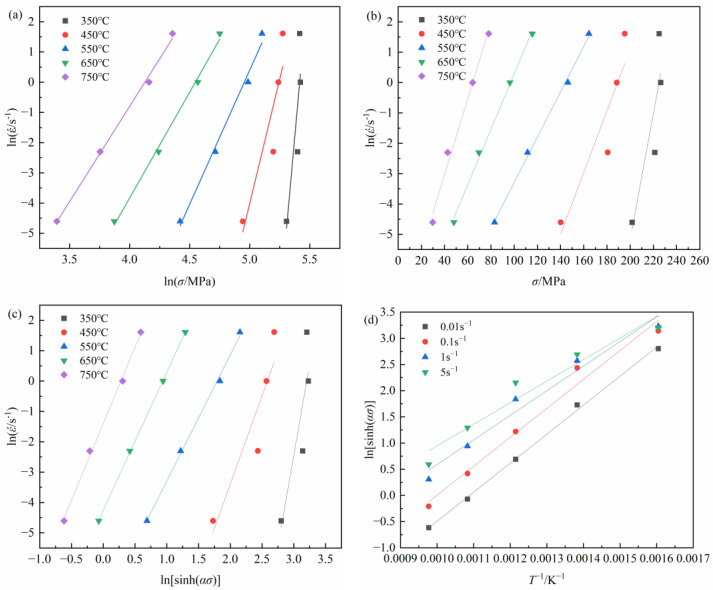
Relationships between: (**a**) ln*έ* and ln*σ*; (**b**) ln*έ* and *σ*; (**c**) ln*έ*-ln[sinh(*ασ*)]; (**d**) 1/*Τ*-ln(sinh(*ασ*)).

**Figure 8 materials-16-03939-f008:**
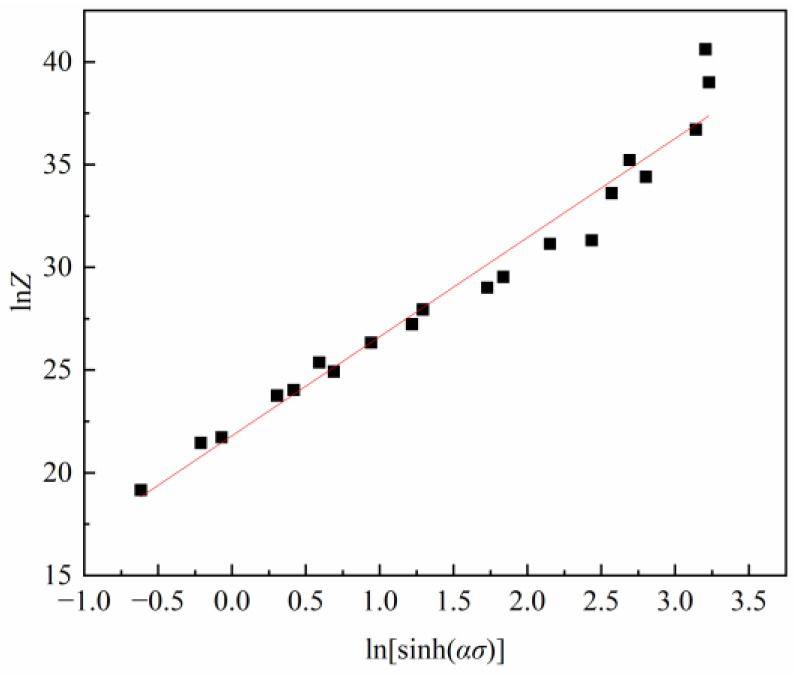
Relationship between ln*Ζ* and ln(sinh(*ασ*)).

**Figure 9 materials-16-03939-f009:**
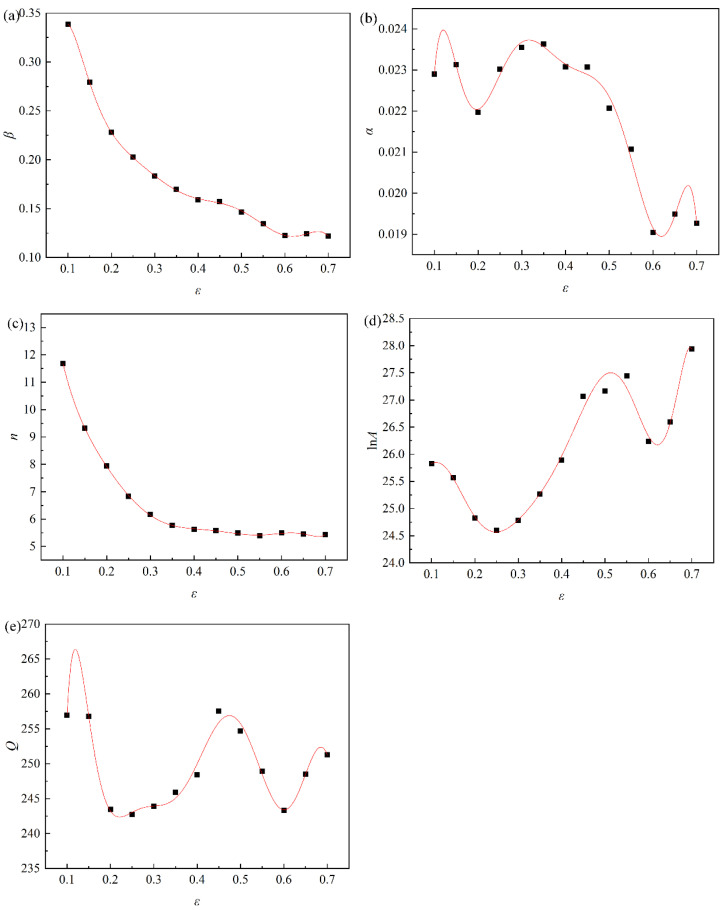
Fitting curves between strain and material constants: (**a**) *β*-*ε*; (**b**) *α*-*ε*; (**c**) *n*-*ε*; (**d**) ln*A*-*ε*; (**e**) *Q*-*ε*.

**Figure 10 materials-16-03939-f010:**
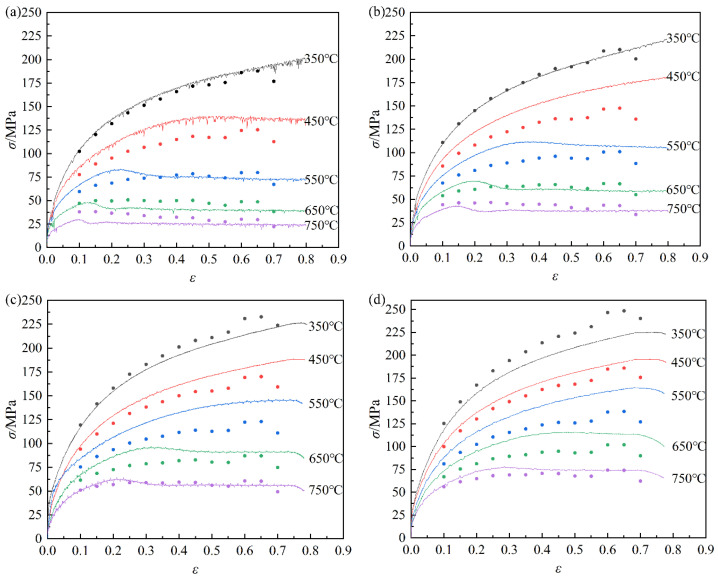
Comparison of pure copper under various deformation circumstances between experimental and predicted values: (**a**) *έ* = 0.01 s^−1^; (**b**) *έ* = 0.1 s^−1^; (**c**) *έ* = 1 s^−1^; (**d**) *έ* = 5 s^−1^.

**Figure 11 materials-16-03939-f011:**
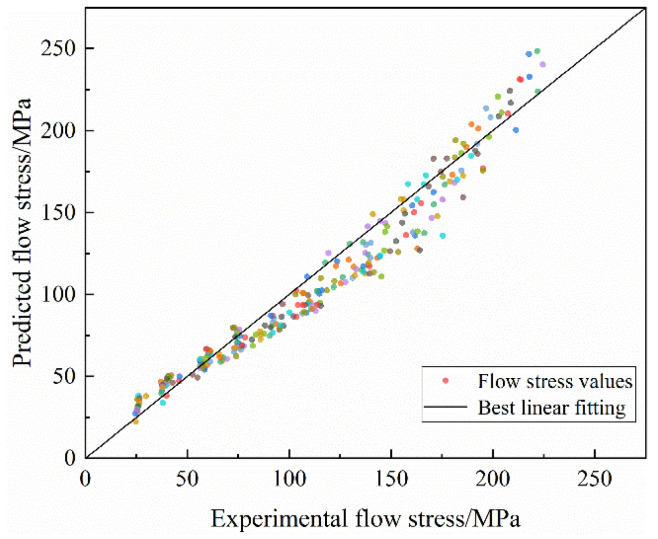
The relevance of flow stress between predicted and experimental values.

**Figure 12 materials-16-03939-f012:**
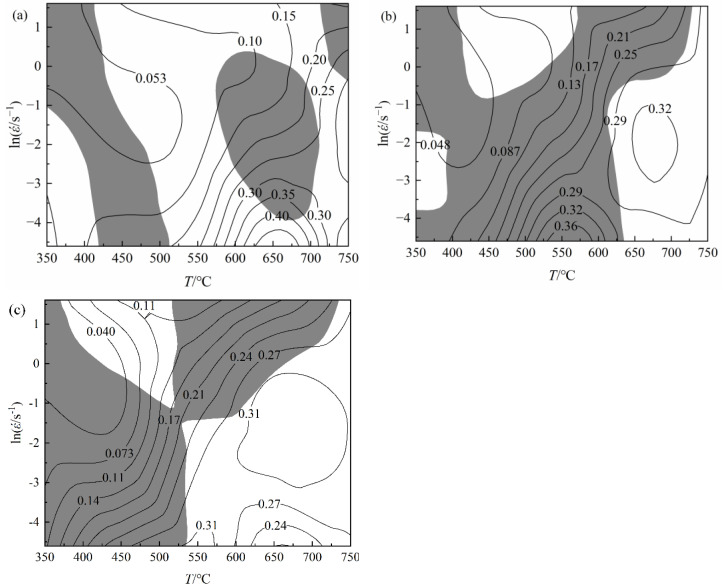
Processing maps of pure copper at different strains: (**a**) *ε* = 0.2; (**b**) *ε* = 0.4; (**c**) *ε* = 0.6.

**Figure 13 materials-16-03939-f013:**
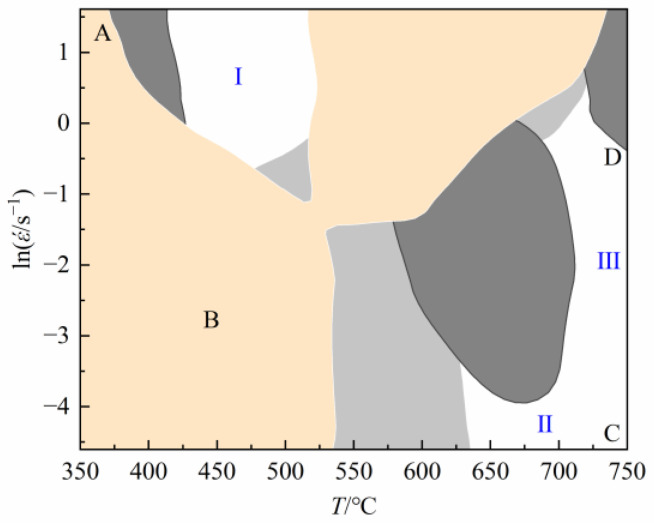
The superposition map of processing maps at strains of 0.2, 0.4, and 0.6.

**Figure 14 materials-16-03939-f014:**
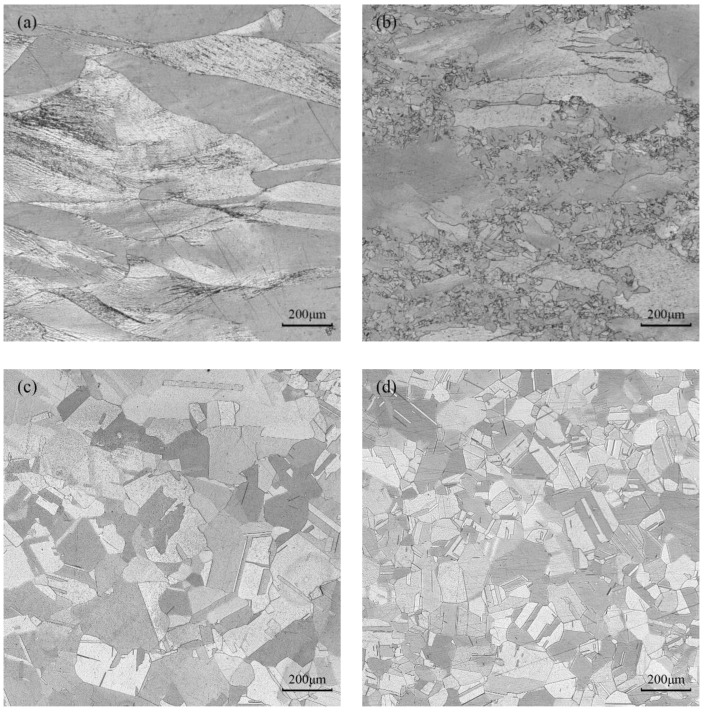
Optical micrographs of pure copper under various compression conditions: (**a**) *T* = 350 °C, *έ* = 5 s^−1^; (**b**) *T* = 450 °C, *έ* = 0.1 s^−1^; (**c**) *T* = 750 °C, *έ* = 0.01 s^−1^; (**d**) *T* = 750 °C, *έ* = 1 s^−1^.

**Table 1 materials-16-03939-t001:** The contents of impurities in copper specimens (wt%).

As	Bi	Fe	Mn	Ni	Pb	P	Sb	Sn	Zn	O
<0.001	<0.001	<0.001	<0.001	<0.001	<0.001	<0.001	<0.001	<0.001	<0.001	0.0021

**Table 2 materials-16-03939-t002:** The value of B for specimens of pure copper under various deformation conditions.

Temp/°C	Strain Rate/s^−1^	*L_f_*/mm	*d_f_*/mm	*B*
350 °C	0.01	7.17	13.92	1.08
	0.1	7.12	14.00	1.07
	1	7.12	13.94	1.08
	5	7.30	13.71	1.09
450 °C	0.01	6.91	14.27	1.07
	0.1	9.95	14.29	1.06
	1	7.21	13.93	1.07
	5	7.21	13.93	1.07
550 °C	0.01	6.82	14.41	1.06
	0.1	6.84	14.44	1.05
	1	7.10	14.18	1.05
	5	6.93	14.29	1.06
650 °C	0.01	6.82	14.31	1.07
	0.1	6.78	14.47	1.06
	1	6.65	14.74	1.04
	5	6.82	14.53	1.04
750 °C	0.01	6.95	14.01	1.10
	0.1	6.85	14.28	1.07
	1	6.96	14.19	1.07
	5	6.70	14.34	1.09

**Table 3 materials-16-03939-t003:** Coefficient values of the fitting relationship between the parameters and the strain.

*i*	*B_i_*	*C_i_*	*D_i_*	*E_i_*	*F_i_*
0	−1.18	−0.144	25.7	−1.67 × 10^3^	41.1
1	52.0	4.99	−2.29 × 10^2^	6.11 × 10^4^	−6.91 × 10^2^
2	−7.04 × 10^2^	−60.0	7.39 × 10^2^	−7.96 × 10^5^	1.25 × 10^4^
3	5.023 × 10^3^	3.81 × 10^2^	6.17 × 10^3^	5.66 × 10^6^	−1.18 × 10^5^
4	−2.15 × 10^4^	−1.41 × 10^3^	−7.35 × 10^4^	−2.44 × 10^7^	6.49 × 10^5^
5	5.78 × 10^4^	3.13 × 10^3^	3.31 × 10^5^	6.66 × 10^7^	−2.20 × 10^6^
6	−9.83 × 10^4^	−4.07 × 10^3^	−8.06 × 10^5^	−1.15 × 10^8^	4.63 × 10^6^
7	1.02 × 10^5^	2.78 × 10^3^	1.12 × 10^6^	1.23 × 10^8^	−5.93 × 10^6^
8	−5.95 × 10^4^	−6.44 × 10^2^	−8.28 × 10^5^	−7.38 × 10^7^	4.21 × 10^6^
9	1.47 × 10^4^	−1.19 × 10^2^	2.55 × 10^5^	1.89 × 10^7^	−1.27 × 10^6^

**Table 4 materials-16-03939-t004:** Range of parameters of unstable regions with different strain conditions.

Strain	Parameters of Unstable Regions	
	Temp/°C	Strain Rate/s^−1^
0.2	350~415	0.09~5
	400~515	0.01~0.09
	570~710	0.02~1.6
	710~750	0.65~5
0.4	350~635	0.03~0.135
	350~415	0.135~5
	415~635	0.135~0.46
	570~730	0.46~5
0.6	350~407	0.01~5
	407~513	0.01~1.39
	513~735	0.206~5

## Data Availability

The authors confirm that the data supporting the findings of this study are available within the article.

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
