# Peer review of "Hot Deformation Behavior and Processing Maps of Pure Copper during Isothermal Compression"

_materials, 2023, doi:10.3390/ma16113939_

Round 1

Reviewer 1 Report

Notes :

1-  The author(s) mentioned in the abstract (Lines No. 21 and 22) that it is suitable deformation temperature at 700-750 oC, while the experimental part did not include a test within a temperature of 700 oC.

2-  The author(s) explained in the abstract that the results of the study showed that with the increase in the deformation temperature, the Flow stress decreases, and this is an axiom.

3-  The author(s)  did not explain on what basis the hold time at 5 min was fixed. before the compression process.

4-  Table No. 3 would have been preferred, including the chemical composition of pure copper, to know the rest of the impurities and not impurities in ppm units.

5-  Dimensions of the compression test sample, mentioning that it was chosen according to the standard.

6-  It was indicated in the experimental part (Line No. 96) that the process of preparing samples for microstructure examination included etching, and no reference was made to the grinding and polishing processes that precede it.

7-  The author(s) (Line No. 96) mentioned the phrase (When the bulge coefficient B is more than 0.9 the results of unidirectional hot compression test are effective), on what basis did the author(s) determine this value of B for this conclusion, the reliable reference is indicated.

8-  The results of the effect of temperature on the true stress-strain curves shown in Figure 3 is axiom.

9-  Pictures of the microstructure examination shown in Figure 4. Note that there is a clear difference in the grain size with the temperature constant at 750 oC, and the difference was in the values of the strain rates. These results need to be further discussed with reference to the Grain Size values for each test.

10- The models proposed by the author(s) that describe the relationship between the variables of the compression process and the behavior and values of the strain rate are good and add a clear research touch to the study.

Reviewer 2 Report

Review of the manuscript "Hot Deformation Behavior and Processing Maps of Pure Copper During Isothermal Compression" by Tiantian Chen, Ming Wen, Hao Cui, Junmei Guo, Chuanjun Wang.

In this work, the behavior of pure copper during hot deformation was studied using isothermal compression tests at various deformation temperatures and deformation rates on the Gleeble-3500 isothermal simulator. The microstructure was analyzed and the microhardness of the samples was measured. Based on the Arrhenius model with deformation compensation, a constitutive equation was established. On the basis of the dynamic model of the material proposed by Prasad, maps of hot working under various loads were obtained.

The result obtained by the authors shows that the flow stress of pure copper decreases with an increase in the strain temperature and a decrease in the strain rate. This is a logical and not a new conclusion.

There are minor remarks:

The initial dimensions of the samples L0 and d0 are not indicated (they are used in the calculation according to equation (1), table 2).

In tables 2 and 4, the abbreviation ‘’Tep’’ is used, it should be corrected to ‘’Temp’’

Rice. 5, а has a different magnification from Fig. 5b–e, which complicates the analysis. For the fig. 14 a similar note.

In general, the results of the work are of practical importance. I believe that the work can be published after minor corrections.

Reviewer 3 Report

The issues, recommendations and questions are as follows.

1. In the line 19-20 (abstract section) be mentioned the percentage of reduction trend.

2. The hardness trend be mentioned in the abstract section.

3. The introduction section is too summary. It is better to add some explanations from similar research work.

4. It is better to extract the tensile strength of samples (according to the Figure 3) and explain the trend of that.

5. The reason for selecting the deformation temperatures be mentioned in line 61-62.

6. The number of used references is too limited. More new references should be added in the manuscript.

It is proper. just 

Reviewer 4 Report

The paper examined the hot deformation behavior of pure copper through isothermal compression tests at various temperatures and strain rates. Metallographic observation and microhardness measurement were carried out and the constitutive equation was established based on the strain compensated Arrhenius model. Hot processing maps were acquired under different strains and the effect of deformation temperature and strain rate on microstructure characteristics was studied. Results show that flow stress decreases with increasing temperature and decreasing strain rate, and the optimal processing parameters for pure copper were determined.

The article is well-structured, with each section presented in a clear and concise manner, effectively conveying relevant results and providing adequate discussion. Considering the significance and potential applications of the research presented in this manuscript, as well as its overall quality, I express my support for its publication in the Journal of Materials. However, I recommend that the questions and comments bellow be thoroughly addressed to further strengthen the manuscript.

1 – The authors should clarify in the text whether the manufacturer supplied the table of impurities present in the raw material listed in table 1, or whether they were measured by the authors. Additionally, it would be helpful if the authors could include the name of the supplier for the copper samples.

2 – I recommend that the organization of Table 1, Figure 1, and Figure 2 be improved in the text as the current layout may be confusing. It would be helpful to present these elements in a more structured and clear manner for the reader's ease of reference.

3 – In line 52, the authors state that "Fig. 1 shows the initial optical micro-structure of pure copper." However, it is important to note that the result presented in Fig. 1 is specific to the pristine sample analyzed and may not represent the general structure of pure copper. Therefore, I recommend that the authors clarify that the result shown is from the pristine sample analyzed, rather than a general result for pure copper.

4 – I recommend that the layout of Table 2 be revised to improve its readability. The current layout may make it difficult for readers to follow the information presented. A clearer and more organized layout could help the reader understand the data more easily.

5 – In figure 10 is presented the comparison between experimental and theoretical curves under different deformation conditions. What catches my attention is a change in the trend of the theoretical curve after the value of 0.7 on the abscissa, which is not always accompanied by the experimental curve. Another point is why theoretical results for values above 0.7 were not presented? I would like to ask the authors to comment on these two points, and if possible, include the theoretical value for at least the value of 0.8.

Moderate revision required.

Round 2

Reviewer 4 Report

The issues raised were satisfactory addressed. In this way I recommend this paper to publication in MDPI Materials.